# Direct assessment of substrate binding to the Neurotransmitter:Sodium Symporter LeuT by solid state NMR

Simon Erlendsson[1,2,3,4,5†], Kamil Gotfryd[3,4,5†‡], Flemming Hofmann Larsen[6], Jonas Sigurd Mortensen[3,4,5], Michel-Andreas Geiger[7], Barth-Jan van Rossum[7], Hartmut Oschkinat[7], Ulrik Gether[3,4,5], Kaare Teilum[1,2*], Claus J Loland[3,4,5*]

[1]Structural Biology and NMR Laboratory, Department of Biology, University of Copenhagen, Copenhagen, Denmark; [2]Linderstrøm-Lang Centre for Protein Science, Department of Biology, University of Copenhagen, Copenhagen, Denmark; [3]Molecular Neuropharmacology Laboratory, Department of Neuroscience and Pharmacology, University of Copenhagen, Copenhagen, Denmark; [4]Lundbeck Foundation Center for Biomembranes in Nanomedicine, University of Copenhagen, Copenhagen, Denmark; [5]Faculty of Health and Medical Sciences, University of Copenhagen, Copenhagen, Denmark; [6]Quality and Technology, Department of Food Science, Faculty of Life Sciences, University of Copenhagen, Copenhagen, Denmark; [7]Leibniz-Institut für Molekulare Pharmakologie FMP, Berlin, Germany

*For correspondence: kaare.
teilum@bio.ku.dk (KT); cllo@sund.
ku.dk (CJL)

[†]These authors contributed equally to this work

**Present address:** [‡]Membrane Protein Structural Biology Group, Department of Biomedical Sciences, Faculty of Health and Medical Sciences, University of Copenhagen, Copenhagen, Denmark

**Competing interests:** The authors declare that no competing interests exist.

**Abstract** The Neurotransmitter:Sodium Symporters (NSSs) represent an important class of proteins mediating sodium-dependent uptake of neurotransmitters from the extracellular space. The substrate binding stoichiometry of the bacterial NSS protein, LeuT, and thus the principal transport mechanism, has been heavily debated. Here we used solid state NMR to specifically characterize the bound leucine ligand and probe the number of binding sites in LeuT. We were able to produce high-quality NMR spectra of substrate bound to microcrystalline LeuT samples and identify one set of sodium-dependent substrate-specific chemical shifts. Furthermore, our data show that the binding site mutants F253A and L400S, which probe the major S1 binding site and the proposed S2 binding site, respectively, retain sodium-dependent substrate binding in the S1 site similar to the wild-type protein. We conclude that under our experimental conditions there is only one detectable leucine molecule bound to LeuT.

## Introduction

The Neurotransmitter:Sodium Symporters (NSSs) are responsible for clearing neurotransmitters, such as dopamine, serotonin, norepinephrine, glycine and GABA from the synaptic cleft. The transporters are thereby crucial for the regulation of synaptic transmission in the CNS and alterations in their function have been linked to several psychiatric and neurological disorders such as depression, bipolar disorders, attention deficit hyperactive disorder (ADHD), epilepsy, and Parkinson's disease (*Broer, 2013*; *Kristensen et al., 2011*). The understanding of the molecular mechanisms and structural (re)arrangements underlying NSS function has advanced significantly in recent years. The most detailed insight into structure-function relationships of NSSs comes from studies of the amino acid transporter, LeuT, from *Aquifex* aeolicus (*Kantcheva et al., 2013*; *Kazmier et al., 2014*; *Malinauskaite et al., 2014*, *2016*; *Piscitelli et al., 2010*; *Quick et al., 2012*; *Shi et al., 2008*; *Singh et al., 2007*; *Wang et al., 2012a*, *2012b*; *Yamashita et al., 2005*). Recent structures of the

**eLife digest** All living cells need amino acids – the building blocks of proteins – in order to survive, yet few cells can make all the amino acids that they need. Instead, transporter proteins in cell membranes must take these molecules from the outside of the cell and release them to the inside. Some cells, including those in the brain, also release amino acids and molecules derived from them into the spaces outside of the cell to send signals to other nearby cells. Again, transporter proteins must move these signaling molecules back inside cells, to stop the signaling and to allow the molecules to be recycled. Importantly, problems with these uptake mechanisms have been linked to disorders such as depression, epilepsy and Parkinson's disease.

One family of transporters involved in the uptake of amino acids are the "Neurotransmitter: Sodium Symporters". Though these proteins are involved in processes that are fundamental to life, it remains unclear exactly how they work. Specifically, it has been heavily debated whether this family of transporters require one or two amino acid molecules to bind at the same time in order to help transport them across the membrane.

Now Erlendsson, Gotfryd et al. have analyzed a bacterial protein in the Neurotransmitter:Sodium Symporter family. This transporter takes up an amino acid called leucine into cells, and is commonly used as a model to understand this family of transporter proteins more generally. Using a technique called solid state nuclear magnetic resonance, Erlendsson, Gotfryd et al. could detect a single molecule of leucine bound to each transporter, but not a second one. This technique could also pinpoint that the leucine was located at the transporter's central binding site. Leucine was never found at the proposed secondary binding site. Together these findings suggest that only one molecule of leucine binds to the transporter at any one time, and that it binds to the transporter's central binding site.

Erlendsson, Gotfryd et al. have shown now how solid state nuclear magnetic resonance can be used to explore in detail how Neurotransmitter:Sodium Symporters move molecules across cell membranes. The next challenge is to use the same experimental setup to characterize other Neurotransmitter:Sodium Symporters. Doing so could potentially lay the groundwork for designing more specific and improved drugs to treat disorders like depression and Parkinson's disease.

drosophila dopamine transporter (dDAT) (*Penmatsa et al., 2013*) and the human serotonin transporter (*Coleman et al., 2016*), which are eukaryotic members of the NSS family, confirm that LeuT is a reliable model protein and proves its value in understanding the molecular function of this class of transporters.

Functional studies of LeuT have suggested the existence of a secondary substrate binding site (S2) located in the extracellular vestibule of LeuT approximately 10 Å from the primary substrate binding site (S1) (*Khelashvili et al., 2013*; *Quick et al., 2009*; *Shi et al., 2008*). The S2 site is suggested to be an allosteric trigger, essential for coupling the energy from the electrochemical gradient to the transport of the solute. The binding of leucine to the S2 site has been measured to have the same affinity (in nM range) as binding to the S1 but does not, as the S1 bound substrate, directly coordinate sodium (*Quick et al., 2012*). However, attempts to crystallize LeuT with substrate bound to the S2 site have so far been unsuccessful, and therefore the existence of the S2 site is supported primarily by radioligand binding assays and guided MD simulations (*Quick et al., 2012*; *Zhao et al., 2011*). Due to the lack of structural evidence, the existence of a high-affinity S2 site has been questioned (*Piscitelli et al., 2010*), supporting the need for employing new techniques for investigating ligand binding in NSS proteins.

Here we investigate the leucine binding properties of LeuT by magic angle spinning (MAS) NMR, aiming at a characterization of the proposed S2 binding site. Our approach offers several advantages: (i) We use microcrystalline samples of LeuT prepared under experimental conditions allowing for conformations capable of ligand binding to both S1 and S2 (*Quick et al., 2012*). (ii) NMR offers information on the full structural ensemble which is unlikely not to include conformers (even lowly populated) prone to bind leucine in S2. (iii) Leucine binding to S1 and S2 may be distinguished by

characteristic chemical shifts that are expected to be different due to different chemical environments, i.e. interacting residues (*Reyes et al., 2011*).

## Results and discussion

Prior to crystallization and NMR experiments we verified the functionality of the produced LeuT wild-type (WT) samples. We initially performed [³H]leucine saturation binding experiments and subsequently assessed Na⁺-dependency of [³H]leucine binding. All experiments were done at a DDM concentration commonly used in *in vitro* assays (i.e., 0.05% corresponding to 5.7x CMC). At this detergent concentration, LeuT was reported to retain binding to both S1 and the putative S2 site (*Quick et al., 2012*). In scintillation proximity binding assays LeuT WT bound [³H]leucine with a dissociation constant ($K_d$) of $12 \pm 1$ nM in the presence of 200 mM sodium. The $EC_{50}$ value calculated for the Na⁺-dependent binding was $47 \pm 4$ mM (*Figure 1—figure supplement 1A–B*). These values are in agreement with those previously reported for LeuT (*Shi et al., 2008*; *Singh et al., 2008*).

As we were primarily interested in a simple readout reporting solely on substrate binding, we purified and kept LeuT in the presence of 1 mM ¹⁵N enriched L-leucine to ensure substrate binding and detection in both sites (*Figure 1A*). With a ¹⁵N natural abundance around 0.3%, the background from the protein amides and amines is sufficiently low to distinguish even weakly populated states originating from the enriched substrate only.

To achieve sufficiently narrow line widths of the NMR signals and to avoid any signal from unbound leucine, we produced microcrystalline samples of LeuT (*Figure 1—figure supplement 2*), and performed cross polarization (CP)-based NMR experiments at temperatures above the freezing point. In all other preparations tested (frozen, lyophilized and proteoliposomes) the signal from the unspecific or unbound leucine completely dominated the spectra (*Figure 1—figure supplement 3A–B*). Using the microcrystalline preparations, we were able to produce the high quality CP-based ¹⁵N detected spectra showing one significant (above 2σ – *Figure 1—figure supplement 4*) peak at 38.2 ppm that could be assigned to the amine of protein bound leucine (*Figure 1B*). In addition to the sharp signal from leucine, much broader signals between 110 and 130 ppm were also observed, which originate from the ¹⁵N natural abundance of the LeuT amides (*Figure 1B*). To further assess whether the intense signal at 38.2 ppm reflects sodium specific leucine binding to LeuT, we performed a parallel experiment substituting Na⁺ with K⁺. Sodium is required for leucine binding (*Zhao et al., 2011*). By the use of ²³Na-NMR we confirmed the presence of only a negligible amount of residual NaCl (at 7.1 ppm), and that no detectable Na⁺ was coordinated in the protein (*Figure 1C*). In the absence of Na⁺, the signal at 38.2 ppm in the ¹⁵N 1D spectrum disappeared as expected for a signal originating from ¹⁵N-leucine bound to LeuT (*Figure 1B*). Worth of note, the amine $NH_3^+$ group of free leucine has a chemical shift of approximately 41 ppm at pH 8 (*Figure 1—figure supplement 5A–B*), demonstrating that the bound substrate resides in a not fully solvent accessible environment. Importantly, we were unable to detect any signal from any additionally bound leucine. Similarly, the ¹³C CP/MAS spectra from the same samples clearly displayed only one single set of sodium dependent leucine signals (*Figure 1—figure supplement 6*).

To investigate whether the origin of the substrate peak at 38.2 ppm was due to leucine binding either to the S1 or the S2 site, we recorded solid state NMR spectra of two variants with compromised leucine binding, F253A (S1) and L400S (S2) (*Figure 2A,B*). For these experiments we lowered the final concentration of the added enriched substrate to 5 μM to ensure proper detimental effect by the mutations. This concentration has previously been reported not to provide any detectable [³H]leucine occupancy in the S2 site of LeuT L400S, but saturated S1 binding (*Quick et al., 2012*). For LeuT WT the specific leucine peak was unaffected by lowering the free leucine concentration (*Figure 2—figure supplement 1*). The F253A mutant has previously been shown to impair binding to the S1 site (*Billesbølle et al., 2015*; *Wang et al., 2012b*). Thus, F253A serves as a S1 disturbing mutant at low substrate concentrations. In the F253A 1D ¹⁵N spectrum, we observed sodium dependent substrate binding with a chemical shift of 38.4 ppm and a slightly lower signal intensity, when compared to the WT spectra (*Figure 2C*). The shift in F253A was consistent for both high (1 mM) and low (5 μM) leucine concentrations (*Figure 2—figure supplement 3*). Most importantly, the chemical shift difference of ~0.2 ppm for the observed bound leucine peak, demonstrates that the ligand is affected by the local environment of the S1 binding site. The L400S mutation was previously suggested to abolish S2 leucine binding (*Quick et al., 2012*). The ¹⁵N spectrum obtained for leucine

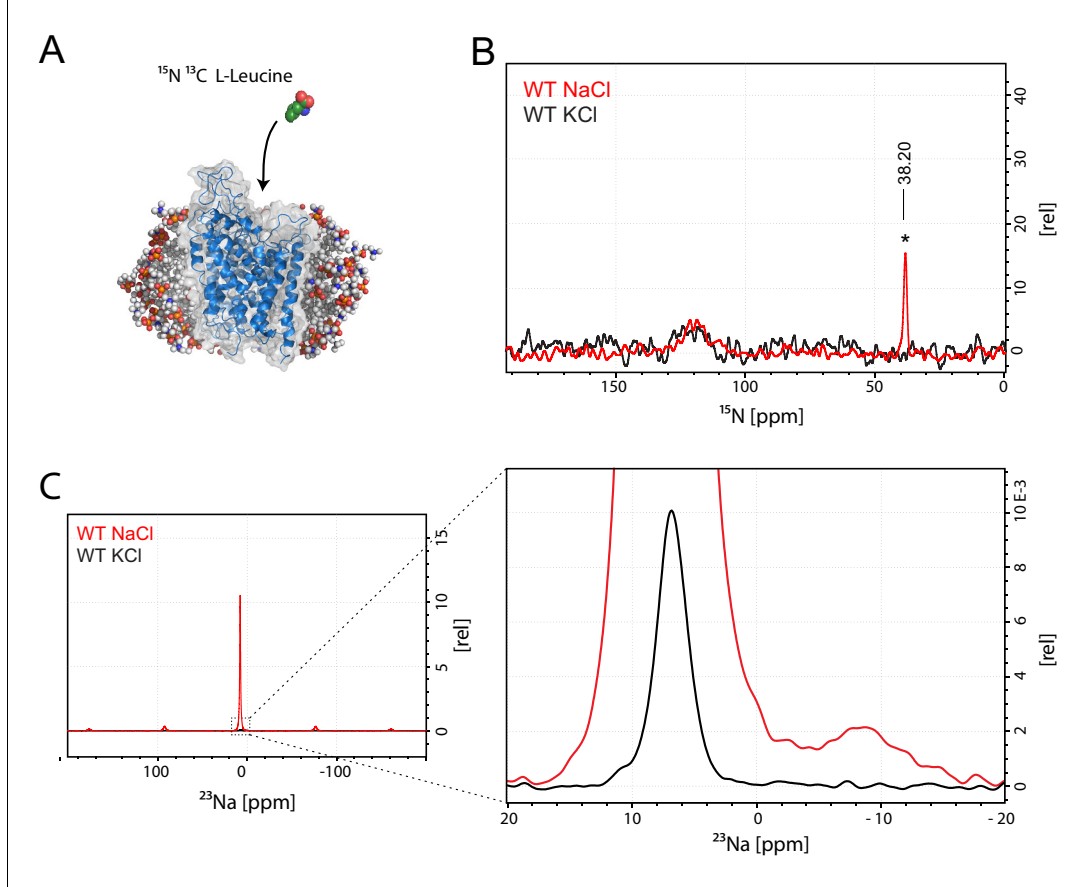

**Figure 1.** Assessment of L-leucine binding to LeuT WT by solid state NMR. (**A**) Cartoon illustration of experimental approach. [15]N enriched L-leucine substrate is added to detergent reconstituted LeuT, which is subsequently crystallized using large scale sitting drop vapour diffusion. Rod-shaped microcrystals form within 24 hr and can be readily harvested. (PDB ID: 3F3E) (**B**) LeuT WT purified in NaCl (red) and LeuT purified in KCl (black). [15]N L-Leucine specific peak is indicated by an asterix with a chemical shift of 38.2 ppm. Spectra are tentatively intensity normalized to the [15]N natural abundance signal from the LeuT backbone amides. Signal-to-noise is calculated to be 21. (**C**) [23]Na-NMR of LeuT WT (red) and LeuT WT in KCl (black) in presence of leucine. Minor peak at −8.9 ppm represents the shape of one or two structural sodium molecules. Despite inequivalent location of the two sodium sites in the LeuT, the coordination mechanism is almost identical which might account for the observation of a single peak in the [23]Na-NMR spectrum instead of two distinct peaks.

The following figure supplements are available for figure 1:

**Figure supplement 1.** Functional characterization of LeuT WT.

**Figure supplement 2.** Microscopy image of LeuT microcrystals.

**Figure supplement 3.** 1D 15N CP/MAS spectrum of frozen and lyophilized LeuT WT samples.

**Figure supplement 4.** 15N L-leucine spectrum substrate peak for LeuT WT.

**Figure supplement 5.** In-solution 1D [15]N spectra of free 98% [15]N L-leucine at different pH.

**Figure supplement 6.** 1D 13C CP/MAS spectrum of microcrystalline LeuT WT samples.

bound to L400S (*Figure 2C*, blue) completely resembled the spectrum obtained with WT protein (*Figure 2C*, red). We were not able to detect any change in intensity or chemical shift. This argues against the possibility that the substrate signal we observe in LeuT WT samples is reporting on a combination of S1 and S2 binding. Also, we reason that it would be highly unlikely for an S1 bound

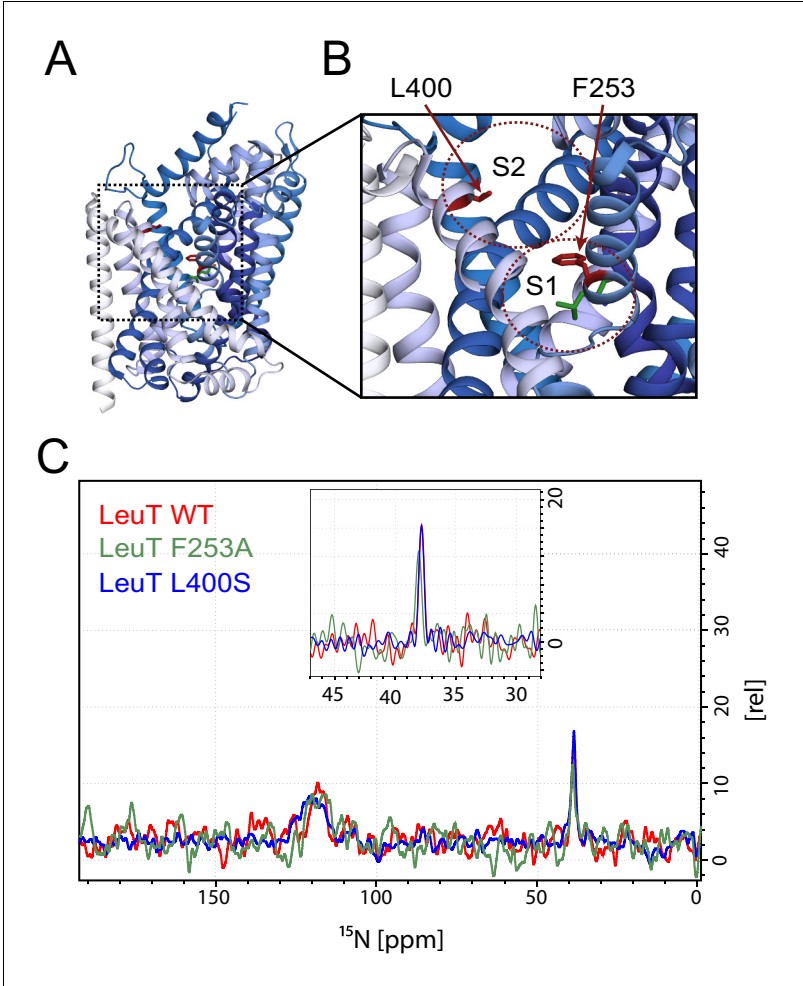

**Figure 2.** Effects of S1 and S2 site mutations on the L-leucine chemical shift . (A–B) Cartoon representation displaying the location of F253 in the S1 site and L400 in the proposed S2 site based on PDB: 3USG (*Wang et al., 2012a*). (C) $^{15}$N 1D NMR spectrum of LeuT WT (red), F253A (green) and L400S (blue). Inset: Close-up of L-leucine specific peak. Spectra are tentatively intensity normalized to the $^{15}$N natural abundance signal from the LeuT amides.

The following figure supplements are available for figure 2:

**Figure supplement 1.** Comparison of spectra derived from LeuT WT purified and crystallized in either 1 mM (red) or 5 uM (purple) free leucine.

**Figure supplement 2.** Power spectra of LeuT WT (red), F253A (green) and L400S (blue).

**Figure supplement 3.** Comparison of LeuT WT and LeuT F253A in the presence of 1 mM free substrate.

**Figure supplement 4.** Cartoon representation of S1 bound substrate (green).

substrate and a putative S2 site-bound substrate to have the same chemical shift as the environment of the putative S2 binding site markedly differs from the S1 binding site. As a major difference, the proposed S2 binding site does not involve direct sodium binding (*Shi et al., 2008*)

In conclusion, although all LeuT samples used in the present study were prepared in DDM at low concentration to exclude previously reported detrimental effects on the S2 binding site, we were only able to identify one single substrate signal at 38.2 ppm in the $^{15}$N spectra and one set of signals in the $^{13}$C spectrum (*Figure 2—figure supplement 2*). We note, however, that at our current signal-

to-noise ratio (~20), we would not be able to detect species populated less than 5% of the structural ensemble. Based on the minor change in chemical shift in the F253A (S1) mutant, and the completely unaltered signal for the L400S (S2) mutant we reason that the observable bound leucine is located at the S1 binding site, thus supporting the idea that LeuT exhibit one single central binding site. We cannot exclude that the detection of S2 binding may only be possible upon the complete transition of the transporter towards a specific (yet unknown) conformation or that unfavourable crystal contacts might complicate S2 binding. However, several crystal structures have shown the binding of antidepressants, which overlaps with the putative S2 site, using these exact conditions (Singh et al., 2007; Zhou et al., 2009). As previously proposed for LeuT (Piscitelli et al., 2010) we speculate that the S2 substrate binding site, if present, is rather a transient site, responsible for optimal functionality of the transporter.

## Materials and methods

### Protein expression and purification

Expression of LeuT WT from *Aquifex aeolicus* was performed according to the protocol described previously (Billesbølle et al., 2015). LeuT WT was expressed in E. coli C41(DE3) transformed with pET16b encoding C-terminally 8xHis-tagged transporter (expression plasmid was kindly provided by Dr E. Gouaux, Vollum Institute, Portland, Oregon, USA). Briefly, isolated bacterial membranes were solubilized in 1% DDM (Anatrace, USA) in the presence of 1 mM 98% $^{15}$N-L-Leucine (Cambridge isotopes, Tewksbury, MA) and the protein was bound to nickel-charged affinity resin (Life Technologies, Carlsbad, CA). Subsequently, protein was eluted in 20 mM Tris-HCl (pH 8.0), 200 mM KCl, 0.05% DDM, 1 mM $^{15}$N-L-Leucine and 300 mM imidazole (KCl sample) or in the same buffer containing NaCl instead of KCl (NaCl sample). LeuT F253A and L400S variants were generated from the *leuT* gene using a QuikChange kit (Agilent Technologies, Santa Clara, CA) and purified similarly to the LeuT WT protein with the difference that 5 µM of $^{13}$C-$^{15}$N-L-Leucine (Cambridge Isotopes) was used for co-purification, and the salt content in all buffers consisted of 50 mM NaCl and 150 KCl. The LeuT F253A variant in the presence of 1 mM substrate was prepared similar to the NaCl sample. Subsequently, all LeuT samples were dialyzed for approx. 36 hr at 4°C in the respective elution buffer without imidazole.

### Functional characterization of LeuT WT

Functional characterization of the LeuT WT purified in KCl was performed using a scintillation proximity assay (SPA) (Quick and Javitch, 2007). Saturation binding of [$^3$H]leucine (50.2 Ci/mmol; PerkinElmer, Waltham, MA) to purified LeuT WT was performed with 100 ng/well (1.66 pmol) of protein in buffer composed of 20 mM Tris-HCl (pH 8.0), 200 mM NaCl, 0.05% DDM, 20% glycerol in the presence of 1.25 mg/ml copper chelate (His-Tag) YSi beads (PerkinElmer). Sodium-dependency was measured at fixed [$^3$H]leucine concentration of 100 nM with increasing concentrations of NaCl (NaCl was substituted with KCl for equal ionic strength) again using 100 ng/well LeuT WT. [$^3$H]Leucine binding was monitored using MicroBeta liquid scintillation counter (PerkinElmer) and data were fitted to a one-site saturation or dose-response function, respectively, using Prism 7 software (Graph-Pad, San Diego, CA).

### Preparation of microcrystals

Microcrystalline samples were produced by large scale sitting drop vapour diffusion method. 1 mL of the protein sample solutions were mixed 1:1 with the crystallization buffer composed of 100 mM NaCl (or KCl), 120 mM MgCl2, 28% PEG400, 100 mM MES or HEPES pH 6.5. The crystallization was carried out at 18°C. After approximately 20 hr a white precipitate could be harvested by centrifugation and transferred directly to the rotor. All samples were freshly prepared immediately before use. Microcrystals where visualized using a Leica M125 microscope with a 1.0x PlanApo objective.

### Lyophilized protein preparation

Protein for lyophilisation was depleted of glycerol during dialysis, snap-frozen in liquid nitrogen and added to a freeze drier. The remaining powder could be transferred directly to the rotor.

## Solution NMR

$^{15}$N-L-Leucine was dissolved to a final concentration of 1 mM in the following buffer: 20 mM Tris-HCl (pH 8.0), 200 mM KCl, 0.05% DDM and added to an Economy WG5 NMR tube. The Experiment was run on a Bruker Avance III 500 MHz operating at a Larmor frequency of 50.667 MHz for $^{15}$N. The directly detected $^{15}$N spectra were recorded using a recovery delay of 2 s and an acquisition time of 500 ms. The total number of 1024 scans were used.

## Solid sate NMR

Microcrystalline LeuT nitrogen spectra were recorded on Bruker Avance III 800 MHz wide bore (89 mm) spectrometer equipped with a 4 mm MAS HCN efree probe. Spectra were obtained at 275 K (measured temperature), at 12500 Hz magic-angle-spinning. $^{15}$N CP/MAS experiments were run for 60 K scans in blocks of 10 K scans and the magnet was fine-tuned between each block. Cross-polarization contact time was set to 1750 us. Initial recovery delay was set to 3 s. Protons were decoupled at 86 kHz during acquisition. $^{13}$C CP/MAS experiments were run for 2 K scans. Cross-polarization contact time was set to 2000 us. The initial recovery delay was set to 3 s. Spectra were displayed using a 1, 10 or 100 Hz line broadening for EM window function in topspin.

Sodium MAS NMR spectra were recorded on a Bruker Avance NMR spectrometer operating at a Larmor frequency of 105.8 MHz for $^{23}$Na using a double resonance probe equipped for 4 mm (o.d.) rotors. All spectra were recorded at room temperature employing a central transition selective 90 degree pulse (1.8 μs), a recycle delay of 2 s, an acquisition time of 40.9 ms, a spectral width of 75.19 kHz and a spin rate of either 9 or 10 kHz. The spectra are referenced to crystalline NaCl at 7.1 ppm.

## Acknowledgement

We thank Trent Franks, Matthias Hiller, Daniel Stöppler for technical assistance with the acquisition of solid state NMR spectra. The work was supported in part by the Danish Independent Research Council – Sapere Aude (0602-02100B) (CJL), The Lundbeck foundation (R108-A10755, R151-2013-14302, R221-2016-847) (CJL, KTE), BioNMR (BIO-NMR-00232) (SE, KTE) and iNEXT (PID 1597) (SE, KTE).

## Additional information

### Funding

| Funder | Grant reference number | Author |
| --- | --- | --- |
| BioNMR | BIO-NMR-00232 | Simon Erlendsson Kaare Teilum |
| iNEXT | PID 1597 | Simon Erlendsson Kaare Teilum |
| Lundbeckfonden | R151-2013-14302 | Kaare Teilum |
| Lundbeckfonden | R221-2016-847 | Kaare Teilum |
| Lundbeckfonden | R108-A10755 | Claus J Loland |
| Det Frie Forskningsråd | Sapere Aude, 0602-02100B | Claus J Loland |
| Det Frie Forskningsråd | DFF-4183-00581 | Claus J Loland |
| bioSYNergy, University of Copenhagen's Excellence Program for Interdisciplinary Research | | Claus J Loland |

The funders had no role in study design, data collection and interpretation, or the decision to submit the work for publication.

### Author contributions

SE, KT, Conception and design, Acquisition of data, Analysis and interpretation of data, Drafting or revising the article; KG, JSM, Acquisition of data, Analysis and interpretation of data, Drafting or

revising the article; FHL, M-AG, B-JvR, Acquisition of data, Drafting or revising the article; HO, Analysis and interpretation of data, Drafting or revising the article; UG, Drafting or revising the article; CJL, Conception and design, Analysis and interpretation of data, Drafting or revising the article

Author ORCIDs

Simon Erlendsson, http://orcid.org/0000-0002-6378-870X

Jonas Sigurd Mortensen, http://orcid.org/0000-0002-6743-276X

Claus J Loland, http://orcid.org/0000-0002-1773-1446

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
