## [Decision Letter]

Thank you for submitting your article "Direct assessment of substrate binding to the Neurotransmitter:Sodium Symporter LeuT by solid state NMR" for consideration by *eLife*. Your article has been reviewed by two peer reviewers, including Baruch Kanner (Reviewer #2), and the evaluation has been overseen by Gary Westbrook as the Senior Editor. The reviewers have discussed the reviews with one another and the Senior Editor has drafted this decision to help you prepare a revised submission.

Summary:

This paper analyzes NMR data on (micro)crystalline preparations of LeuT and two mutants in reference to previous structural and functional studies that led to conflicting views on the number of leucine binding sites: a central site (S1) on which there is general agreement and a potential additional site (S2) located in the external vestibule of the transporter. This S2 site has been proposed to play an important role in the transport mechanism. Clearly, a detailed study of ligand binding to membrane proteins is a central aspect for understanding their workings in cell membranes. The LeuT system has served as a powerful bacterial model system to understand the details of ligand and ion concentrations as well as the role of lipids and detergents. The crystal structures of the bacterial amino acid transporter LeuT have been determined in various conformations. It has been established that LeuT is a very useful model for Neurotransmitter:Sodium:Symporters. For these reasons, this work addresses an important, so far unresolved issue.

In this manuscript, the issue is addressed using a novel approach with solid state NMR to study leucine binding in the wild type and two mutant transporters. The latter are expected to perturb either of the proposed sites. The authors describe a single sodium dependent signal in the wild type with similar chemical shifts and magnitudes at high (1mM) or low (5μM) substrate concentrations. At the low concentration, an identical signal was observed with the S2 site mutant, whereas the amplitude of the signal of the S1 mutant was lowered and the shift was different. The reviewers were interested in the topic and approach, but raised several issues that will need to be addressed in a revised manuscript.

Essential revisions:

1) Reliability of the NMR data interpretation.

The main conclusion of this work, i.e., the presence of only one bound Leucine site, is largely based on one-dimensional NMR spectra of rather limited signal to noise. This is especially true for Figure 2 where, considering the base line, the signal to noise S/N is roughly 4:1 (and not as claimed in Figure 1, 21:1). Although the data shown in Figure 2 seem to have been recorded with similar acquisition times, the signal to noise seems to vary significantly with the best S/N for the L400S mutant.

As a result, the "tentative normalization" of the 38 ppm peak to the NH backbone signals relative to the 38.2 ppm peak is questionable and the signal modulation at the 38 ppm resonance for the 3 samples is comparable to the noise level. Why did the authors not conduct longer experiments to obtain a better signal to noise ratio? In addition, the 15N signal of the F253A mutant refers to the spectrum with the lowest signal to noise and the claimed 0.2 ppm peak shift could easily disappear with a slight change in phase correction. Even if this shift is really present, it is probably smaller than the intrinsic NMR line width.

2) General relevance and implications in reference to previous work.

MAS-NMR has been used to study proteins embedded in proteoliposomes for decades and it remains unclear why such experiments were not conducted here. Because the substrate remains labeled, they should be readily possible. To the reviewer, such data would greatly improve the general relevance of this study because they would allow to ultimately compare detergent and lipid bilayer data on the atomic level (see Quick et al., 2012).

3) Because in the literature the S1 mutant has been described as capable of binding the substrate, but with lower affinity, the authors attribute the lowered amplitude to the lowered substrate affinity but do not substantiate this claim by measuring the signal at the high substrate concentration. It is essential to do this measurement and it will be important to see if the chemical shift changes or not.

4) In the last sentence the authors unnecessarily soften up their conclusion, probably to try to be "politically correct". Assuming that the suggested experiment strengthens their conclusion, the sentence could read something like "Our data, using a novel approach to determine substrate binding, support the idea that LeuT exhibits a single central binding site".

---

## [Author Response]

*Essential revisions:*

1) Reliability of the NMR data interpretation.

*The main conclusion of this work, i.e., the presence of only one bound Leucine site, is largely based on one-dimensional NMR spectra of rather limited signal to noise. This is especially true for Figure 2 where, considering the base line, the signal to noise S/N is roughly 4:1 (and not as claimed in Figure 1, 21:1). Although the data shown in Figure 2 seem to have been recorded with similar acquisition times, the signal to noise seems to vary significantly with the best S/N for the L400S mutant.*

*As a result, the "tentative normalization" of the 38 ppm peak to the NH backbone signals relative to the 38.2 ppm peak is questionable and the signal modulation at the 38 ppm resonance for the 3 samples is comparable to the noise level. Why did the authors not conduct longer experiments to obtain a better signal to noise ratio? In addition, the 15N signal of the F253A mutant refers to the spectrum with the lowest signal to noise and the claimed 0.2 ppm peak shift could easily disappear with a slight change in phase correction. Even if this shift is really present, it is probably smaller than the intrinsic NMR line width.*

We fully acknowledge that the intensities of the peaks in our spectra are low. Although it would be very nice with better signal-to-noise, we have already pushed the amount of NMR time used for recording a single 1D spectrum to the limit. The spectra presented in the paper were signal averaged in 6 blocks of 60K scans. The magnet was re-tuned before each block of scans, which resulted in just about 64 hours for each 1D experiment. Consequently, increasing the signal to noise by a factor of two would require at least 256 hours of experiment time, which in our opinion would be too much. It could risk a significant destabilization of the sample.

The data presented in Figure 1 clearly demonstrate the absence of additional Leucine binding sites that are populated more than 5% relative to the S1 site and that have a chemical shift difference of more than 30Hz (0.04ppm) compared to the S1 site.

In Figure 2, spectra of the variants F253A and L400S are shown. We did not obtain as much crystallized protein for these variants as for LeuT WT and the signal-to-noise for the S1 peaks in these spectra are indeed lower than for the WT spectrum (4:1 and 11:1, respectively). However, the peaks are sharp and their positions are well defined (intrinsic line widths of ~30 Hz). We agree with the reviewers that our presentation of the data can be improved. In the original figure we had processed the data with a line broadening of 100 Hz, hiding the resolution. To accommodate this issue, we have presented the data in the inset of Figure 2 with only 1 Hz line broadening. In addition, in Figure 2—figure supplement 2 we have produced power spectra, and aligned the intensities of the three constructs to demonstrate that the 0.2 ppm difference cannot be accounted for by minor phasing errors.

2) General relevance and implications in reference to previous work.

*MAS-NMR has been used to study proteins embedded in proteoliposomes for decades and it remains unclear why such experiments were not conducted here. Because the substrate remains labeled, they should be readily possible. To the reviewer, such data would greatly improve the general relevance of this study because they would allow to ultimately compare detergent and lipid bilayer data on the atomic level (see Quick et al., 2012).*

Proteoliposome preparations are very suitable for most MAS-NMR membrane protein studies, and we did indeed try this approach for the substrate binding but without any luck. As demonstrated in Figure 1—figure supplement 3, having either frozen or completely solid samples will cause the free leucine to dominate the spectrum completely. When recording proteoliposome data at room temperature we do not see any signal in the CP based experiments from substrate, which might be an effect of large flexibility or too low sensitivity (lipids will constitute most of the material in the ssNMR rotor in order to preserve the functionality of the transporter).

In the microcrystalline samples we have no unbound Leucine present as no signals are observed in the potassium purified protein, which serves as our negative control. Finally, microcrystalline samples have greatly improved resolution (more order) compared to fluid detergent/lipid preparations, which is crucial for our conclusions. To clarify these points, we have rephrased the text which now reads:

“To achieve sufficiently narrow line widths of the NMR signals and to avoid any signal from unbound leucine, we produced microcrystalline samples of LeuT (Figure 1—figure supplement 2), and performed cross polarization (CP)-based NMR experiments at temperatures above the freezing point. In all other preparations tested (frozen, lyophilized and frozen proteoliposomes) the signal from the unbound leucine completely dominated the spectra (Figure 1—figure supplement 3).”

*3) Because in the literature the S1 mutant has been described as capable of binding the substrate, but with lower affinity, the authors attribute the lowered amplitude to the lowered substrate affinity but do not substantiate this claim by measuring the signal at the high substrate concentration. It is essential to do this measurement and it will be important to see if the chemical shift changes or not.*

We thank the reviewers for this insightful suggestion. As proposed we have performed the experiment for F253A at 1 mM free substrate concentration (Figure 2—figure supplement 3). To this end, we realize that the fourth paragraph of the Results and Discussion puts unnecessary emphasis on the slight change in intensity. Our conclusion that the bound leucine senses changes in the S1 local environment (caused by the F253A mutation) and therefore are bound in S1, rests primarily on the differences in chemical shift of the leucine signals between LeuT WT and F253A. The comparison of the signal intensities relies on tentative normalization based on the natural abundance resonances, which are not suitable for making conclusions on subtle differences in the affinity, as also noted above.

Taken together, we have changed the paragraph to read:

“In the F253A 1D ^15^N spectrum, we observed sodium dependent substrate binding with a chemical shift of 38.4 ppm and a slightly lower signal intensity, when compared to the WT spectra (Figure 2). The shift in F253A was consistent for both high (1 mM) and low (5 µM) leucine concentrations (Figure 2—figure supplement 3). Most importantly, the chemical shift difference of ~0.2 ppm for the observed bound leucine peak, demonstrates that the ligand is affected by the local environment of the S1 binding site.”

*4) In the last sentence the authors unnecessarily soften up their conclusion, probably to try to be "politically correct". Assuming that the suggested experiment strengthens their conclusion, the sentence could read something like "Our data, using a novel approach to determine substrate binding, support the idea that LeuT exhibits a single central binding site".*

The last paragraph of the Results and Discussion now reads,

“Based on the minor change in chemical shift in the F253A (S1) mutant, and the completely unaltered signal for the L400S (S2) mutant we reason that the observable bound leucine is located at the S1 binding site, thus supporting the idea that LeuT exhibit one single central binding site.”